

# Small extracellular vesicles derived from umbilical cord mesenchymal stem cells repair blood-spinal cord barrier disruption after spinal cord injury through down-regulation of Endothelin-1 in rats

Chenhui Xue[1], Xun Ma[1,2], Xiaoming Guan[1,2], Haoyu Feng[1,2], Mingkui Zheng[1] and Xihua Yang[3]

[1] Third Hospital of Shanxi Medical University, Shanxi Bethune Hospital, Shanxi Academy of Medical Sciences, Tongji Shanxi Hospital, Taiyuan, Shanxi, China
[2] Department of Orthopedics, Shanxi Bethune Hospital, Shanxi Academy of Medical Sciences, Tongji Shanxi Hospital, Third Hospital of Shanxi Medical University, Taiyuan, Shanxi, China
[3] Laboratory Animal Center, Shanxi Province Cancer Hospital/Shanxi Hospital Affiliated to Cancer Hospital, Chinese Academy of Medical Sciences/Cancer Hospital Affiliated to Shanxi Medical University, Taiyuan, Shanxi, China

Corresponding authors
Xun Ma, maxun2532@sina.com
Xiaoming Guan,
doctor_gxm2012@126.com

## ABSTRACT

Spinal cord injury could cause irreversible neurological dysfunction by destroying the blood-spinal cord barrier (BSCB) and allowing blood cells like neutrophils and macrophages to infiltrate the spinal cord. Small extracellular vesicles (sEVs) derived from mesenchymal stem cells (MSCs) found in the human umbilical cord have emerged as a potential therapeutic alternative to cell-based treatments. This study aimed to investigate the mechanism underlying the alterations in the BSCB permeability by human umbilical cord MSC-derived sEVs (hUC-MSCs-sEVs) after SCI. First, we used hUC-MSCs-sEVs to treat SCI rat models, demonstrating their ability to inhibit BSCB permeability damage, improve neurological repair, and reduce SCI-induced upregulation of prepro-endothelin-1 (prepro-ET-1) mRNA and endothelin-1 (ET-1) peptide expression. Subsequently, we confirmed that hUC-MSCs-sEVs could alleviate cell junction destruction and downregulate MMP-2 and MMP-9 expression after SCI, contributing to BSCB repair through ET-1 inhibition. Finally, we established an *in vitro* model of BSCB using human brain microvascular endothelial cells and verified that hUC-MSCs-sEVs could increase the expression of junction proteins in endothelial cells after oxygen-glucose deprivation by ET-1 downregulation. This study indicates that hUC-MSCs-sEVs could help maintain BSCB's structural integrity and promote functional recovery by suppressing ET-1 expression.

## INTRODUCTION

Spinal cord injury (SCI), a central nervous system (CNS) disorder, is characterized by persistent sensory and motor abnormalities. Due to its high morbidity and death, SCI poses a global health burden (*Simpson et al., 2012*; *Ahuja et al., 2017*). Injury to neurons and axons directly causes the majority of primary spinal cord injuries, whereas "spinal cord microenvironment imbalance" is thought to be the cause of secondary spinal cord injury exacerbation (*Fan et al., 2018*). Restoration of neurological function following SCI is dependent on the integrity of the blood-spinal cord barrier (BSCB), which consists of continuous endothelial cells joined by molecular junctions. The BSCB also serves as a barrier to prevent paracellular and transcellular movement (*Jin et al., 2021*). Additionally, it may control and limit the entry of outside chemicals into the CNS, maintain the microenvironment's homeostasis, and significantly influence the pathophysiological process of various neurological illnesses (*Bartanusz et al., 2011*).

Clinical management for spinal cord injury extends patient lifespan but has limited impact on nerve recovery. And stem cell transplantation is a promising avenue for treating SCI. In this context, human umbilical cord mesenchymal stem cells (hUC-MSCs) have emerged as a promising source of human stem cells for therapeutic interventions due to their widespread availability and minimal ethical concerns (*Ullah, Subbarao & Rho, 2015*). Transplantation of hUC-MSCs into traumatic spinal cord injuries has shown neuro regenerative properties and improved functional outcomes (*Gao et al., 2020*). However, similar to other cell-based therapies, hUC-MSC transplantation carries potential risks, including infections, embolism, acute immunogenicity, chronic immunogenicity, and tumorigenicity (*Saeedi, Halabian & Imani Fooladi, 2019*).

To address safety concerns associated with the administration of living cells, cell-free therapy has emerged as a compelling alternative. Being acknowledged as vital facilitators of intercellular communication, naturally secreted extracellular vesicles (EVs) possess the innate capacity to stimulate tissue regeneration and, therefore, hold promise as biotherapeutic agents (*Kim et al., 2022*). sEVs are small EVs ranging from 30 to 100 nm produced by all cells (*Zhang et al., 2019*). They are essential components of paracrine secretions and mediate cell-to-cell communication by transferring genetic material signals, such as non-coding RNAs and mRNAs, as well as proteins, and inhibiting their degradation (*Hessvik & Llorente, 2018*). Stem cell-derived sEVs exhibit therapeutic benefits comparable to stem cell transplantation through mechanisms such as anti-apoptosis, immunomodulation, anti-inflammatory effects, and the promotion of angiogenesis (*Han et al., 2016*).

In the current context, the restoration of neurological function following spinal cord injury relies on the integrity of the blood-spinal cord barrier (BSCB), although the precise regulatory mechanisms remain unclear. Endothelin (ET) is a potent vasoconstrictor involved in various responses associated with CNS disorders (*Leslie et al., 2004*; *Hostenbach et al., 2016*). Endothelin-1 (ET-1) levels have been found to increase in the cerebrum tissue of traumatic brain injury (TBI) models and spinal cord tissue of SCI models (*Maier et al., 2007*; *A et al., 2019*; *Michinaga et al., 2020*). Overexpression of ET-1 causes loss of

endothelial integrity, increases blood–brain barrier permeability, aggravates ischemia and hypoxia, as well as induces tissue necrosis and apoptosis (*Peters et al., 2003*).

In summary, SCI presently has limited effective therapeutic options due to its refractory nature. Human umbilical cord mesenchymal stem cell-derived small extracellular vesicles (hUC-MSCs-sEVs) offers a promising approach to SCI treatment (*Kang & Guo, 2022*). However, their effectiveness and underlying mechanisms remain incompletely understood. Therefore, in this study, we investigated the function of hUC-MSCs-sEVs in BSCB repair after SCI, focusing on the involvement of ET-1.

## MATERIALS & METHODS

### Cell culture

The HUC-MSCs were purchased from Fuyuan Biotechnology (Fuyuan Biotechnology Co., Ltd. Shanghai, China). Cells were cultured in Dulbecco's modified Eagle's medium (DMEM, Gibco, NY, USA) supplemented with 10% fetal bovine serum (FBS, Gibco, NY, USA) and 1% penicillin/streptomycin (Thermo Fisher Scientific, Waltham, MA, USA). Human brain microvascular endothelial cells (HBMECs) were purchased from Meisen Cell Technology (Meisen Cell Technology, Zhejiang, China). Cells were cultured in an endothelial cell medium (ScienCell Research Laboratories, San Diego, CA, USA) and incubated in a humidified atmosphere at 5% $CO_2$ and 37 °C.

### sEVs isolation and characterization

We adhered to the guidelines for the isolation, characterization, and functional analysis of EVs as stipulated in a consensus document published by the International Society of Extracellular Vesicles (*Koeppen et al., 2021*). Extracellular vesicles were harvested from passage (P) 3 to P5 of the hUC-MSCs. To isolate sEVs, the hUMSC-conditioned medium was first centrifuged at 500 g for 10 min to remove cells. Subsequently, the supernatant was centrifuged at 10,000 g for 30 min to eliminate apoptotic vesicles and other debris. The resulting liquid was then filtered through a 0.22 mm filter. The sEVs were then collected as a pellet using ultracentrifugation (Beckman Optima XPN, 45Ti) at 110,000 g for 70 min. The sEVs pellet was resuspended in phosphate-buffered saline (PBS) for purification and subjected to another round of ultracentrifugation at 110,000 g for 70 min to remove the contaminating proteins. Finally, the sEVs were resuspended in PBS. The Pierce BCA Protein Assay Kit (Thermo Fisher Scientific, Waltham, MA, USA) was used to assess the protein content of the sEVs. The hUC-MSCs-sEVs sample was stored at −80 °C for further analysis. The size of sEVs was determined by nanoparticle tracking analysis (NTA) using ZetaView S/N 17-310 (Particle Metrix, Meerbusch, Germany) along with its associated software. Additionally, transmission electron microscopy (TEM; JEOL Ltd., Tokyo, Japan) was used to morphologically examine isolated sEVs. Western blot analysis was utilized to determine the levels of CD63 (Abcam, Cambridge, UK) and TSG101 (Abcam, Cambridge, UK) in sEVs.

### Experimental animals

All animal experimental protocols conformed to the Guide for the Care and Use of Laboratory Animals from the National Institutes of Health (NIH Publications (8023),

and all procedures were approved by the Shanxi Provincial People's Hospital Institutional Animal Care and Use Committee (Approval No. (2022-089). Adult female Sprague–Dawley rats weighing between 220 and 250 g were obtained from the Laboratory Animal Center of Shanxi Cancer Institute (animal production certificate # SCXK (Jin) 2017-0001; Shanxi, China). The entire experimental process was conducted at the same institution (animal usage certificate # SYXK (Jin) 2017-0003; Shanxi, China). Rats were housed in pathogen-free environments, with two to three animals per cage. They were provided with a standard commercial diet, had ad libitum access to water, and maintained under control humidity (40–60%) in a 12-hour light-dark cycle. After the assay, all the surviving animals were euthanized by an intraperitoneal injection of barbiturates at the Laboratory Animal Center of Shanxi Cancer Institute. The rats were randomly assigned to the following four groups: control ($n = 20$), SCI ($n = 20$), exo ($n = 20$), and ET-1 ($n = 20$) groups.

## Spinal cord injury and treatment

Rats were anesthetized with 1% sodium pentobarbital (3 mL/kg, i.p.), and a median dorsal incision was performed at the T10 segment. The surrounding tissues were carefully dissected to expose the T10 vertebral body, spinous process, and spinal cord. The muscles were dissected layer by layer while preserving the integrity of the dura mater. In the sham-operated group, the wound was sutured layer by layer after sterilization. A spinal cord injury model was established using the modified Allen's method for the remaining three groups. The spinal cord injury was created at the T10 level using a standardized force (10 g × 5 cm), and the successful modeling was confirmed by the presence of congestion, edema, double hind limb convulsions, and spastic tail swing at the site of injury in the spinal cord.

## Treatment

Postoperatively, the ET-1 group was injected 10 µL (1 µg/mL) of ET-1 directly into the injured site using a microinjector. The wound was subsequently sutured layer by layer after rinsing with saline and disinfection. The Sham and SCI groups were administered 200 µL PBS solution in the tail vein immediately after injury and 1 and 2 days post-injury, while 40 µg sEVs (200 µg/mL) solution was administered to the sEVs and ET-1 groups. This dosage was chosen based on previous study with sEVs in rats (*Zhang et al., 2015*). All rats were given intraperitoneal penicillin (200,000 U/d) for 3 days, while their bladder was manually massaged twice or thrice daily to facilitate urination until the urinary function was restored.

## Behavioral tests

The Basso, Beattie, and Bresnahan (BBB) ratings were used to assess the functional impairments following SCI. Two independent examiners blinded to the experimental groups evaluated the BBB scores on an open-field scale. Evaluations were performed at 1, 3, 5, 7, 14, and 21 days after surgery to monitor the progression of functional recovery in the rats.

## Western blot analysis

For western blot analysis, spinal cord tissue was mixed with RIPA lysate, lysed on ice for 30 min, and then centrifuged at 15,000× g for 10 min at 4 °C. For protein analysis *in vitro*, HBMECs were lysed in RIPA buffer with protease and phosphatase inhibitors. The protein content of the supernatants was quantified using the PierceTM BCA protein assay kit (Thermo Fisher Scientific, Waltham, MA, USA), and the supernatant was collected for subsequent protein analysis. A 10% gel was used to separate equivalent quantities of 20 mg of protein, which were then transferred onto a polyvinylidene difluoride (PVDF) membrane (Merck Millipore, Darmstadt, Germany). Following blocking with 5% nonfat milk in TBS with 0.05% Tween 20 for 1 h, the membrane was incubated with primary antibodies against ZO-1 (61-7300; Thermo Fisher Scientific, Waltham, MA, USA), beta-catenin (ab32572, Abcam, Cambridge, UK), occluding (ab216327, Abcam, Cambridge, MA, USA), claudin-5 (352500; Santa Cruz Biotechnology, Inc., Dallas, TX, USA), MMP-2 (ab92536, Abcam, Cambridge, MA, USA), and MMP-9 (ab76003, Abcam, Cambridge, MA, USA) at 4 °C overnight (around 20 h). After three TBS-T washes, the membranes were incubated with the secondary antibodies for 1 h at room temperature. The protein bands were visualized using an automated gel imaging system (Bio-Rad ChemiDoc MP, Bio-Rad, Hercules, CA, USA), while the band densities were measured using ImageJ software. The relative density ratios normalized to the Sham or Control group were used to describe the findings.

## Real-Time PCR

Spinal cord tissue samples were subjected to isopropanol precipitations after total RNA extraction using the RNAiso plus protocol (TaKaRa Bio Inc., Shiga, Japan). The extracted total RNA was used to synthesize first-strand cDNA. Quantitative reverse transcription-polymerase chain reaction (RT-PCR) was used to measure mRNA levels using SYBR Green fluorescent probes. The SYBR Green Master Mix (TaKaRa Bio Inc., Shiga, Japan) was added to each reverse-transcription product, and the reaction mixture was then subjected to amplification using a CFX96 Touch Real-Time PCR Detection System (Bio-Rad, Hercules, CA, USA).

The following primer pairs were used for amplification:

Prepro-ET-1 Forward: 5′-GTGAGAACGGCGGGGAGAAAC-3′

Reverse: 5′- AATGATGTCCAGGTGGCAGAAGTAG -3′

GAPDH: Forward: 5′-CTCTGATTTGGTCGTATTGGG-3′

Reverse: 5′-TGGAAGATGGTGATGGGA TT-3′

Serial dilutions of each amplicon were also amplified to generate standard curves for the quantification of the PCR products. The copy numbers of each PCR product, equal to 1 µg of total RNA, was used to calculate the quantity of mRNA. The prepro-ET-1 mRNA expression levels were normalized to GAPDH values.

## Evans blue dye assays

Evans Blue Dye Assays were performed to assess BSCB permeability. A 2% Evans blue saline solution (two mL/kg) was administered into the tail vein of rats 7 days after SCI. After 2 h, the rats were anesthetized with 1% sodium pentobarbital (three mL/kg), and saline was

perfused through the heart until clear fluid began to flow from the right atrium. A 1 cm segment of the injured spinal cord, centered around the injury site, was carefully dissected, weighed, and homogenized in a 50% trichloroacetic acid solution. The homogenate was then centrifuged at 10,000 g for 10 min, and the supernatant was collected. The absorbance of the sample was measured using a spectrophotometer (with an excitation wavelength of 620 nm and an emission wavelength of 680 nm). The established standard curve was used to determine the quantity of Evans dye present in the tissue ($\mu$g/g).

### FITC- dextran assays
The rats received an intravenous injection of 2% FITC-dextran (MW 70 kDa, 4 mg/kg; Sigma-Aldrich, Burlington, MA, USA) solution in PBS *via* the tail vein 1 day after SCI. After 2 h, the rats were injected with 10% chloral hydrate, followed by perfusion with 0.9% normal saline. The FITC-dextran-damaged spinal cord tissues were weighed, homogenized in PBS, and centrifuged. The optical density of the supernatant was measured using a spectrophotometer at an excitation wavelength of 493 nm and an emission wavelength of 517 nm to assess the presence of FITC-dextran.

### Oxygen–glucose deprivation/reoxygenation procedure
Oxygen–glucose deprivation/reoxygenation (OGD/R) procedures were conducted following previously established protocols (*Sun et al., 2017*). Briefly, cultivated HBMECs were washed thrice with PBS and then transferred to serum-free DMEM without glucose (Gibco, Life Technologies, USA). Subsequently, the HBMECs were subjected to oxygen-glucose deprivation (OGD) by placing them in an anaerobic chamber containing 1% $O_2$, 5% $CO_2$, and 94% $N_2$ at 37 °C for 6 h. After being exposed to OGD for 6 h, the HBMECs were washed once with PBS and then incubated under normal conditions (reoxygenation) for 24 h.

### Cell viability assay
Cell counting kit-8 (CCK-8) assay was used to assess cell viability. HBMECs were cultured in endothelial cell media and seeded in 96-well plates. After 1, 2, 3, 4, and 5 days of incubation, 10 $\mu$L of CCK-8 reagent (Dojindo, Japan) was added to the culture medium. A microplate reader (Bio-Rad 680, Hercules, CA, USA) was then used to measure the absorbance of each well at 450 nm.

### Paracellular permeability assay
HBMECs were seeded overnight in a 200- $\mu$L medium at a density of $1 \times 10^5$ cells/well on Transwell permeable supports (PET membrane 24-well cell culture inserts with 0.4-$\mu$m pore size; Corelle; Corning Life Sciences, Corning, NY, USA). Subsequently, the cells were subjected to OGD for 6 h, followed by reoxygenation for 22 h (OGD6h/R22h). The cells were then exposed to media containing FITC-dextran (1 mg/ml) for 2 h. The amount of FITC-dextran passing through the Transwell (in the lower chambers) was determined using an enzyme-labeled meter with an excitation wavelength of 493 nm and an emission wavelength of 517 nm.

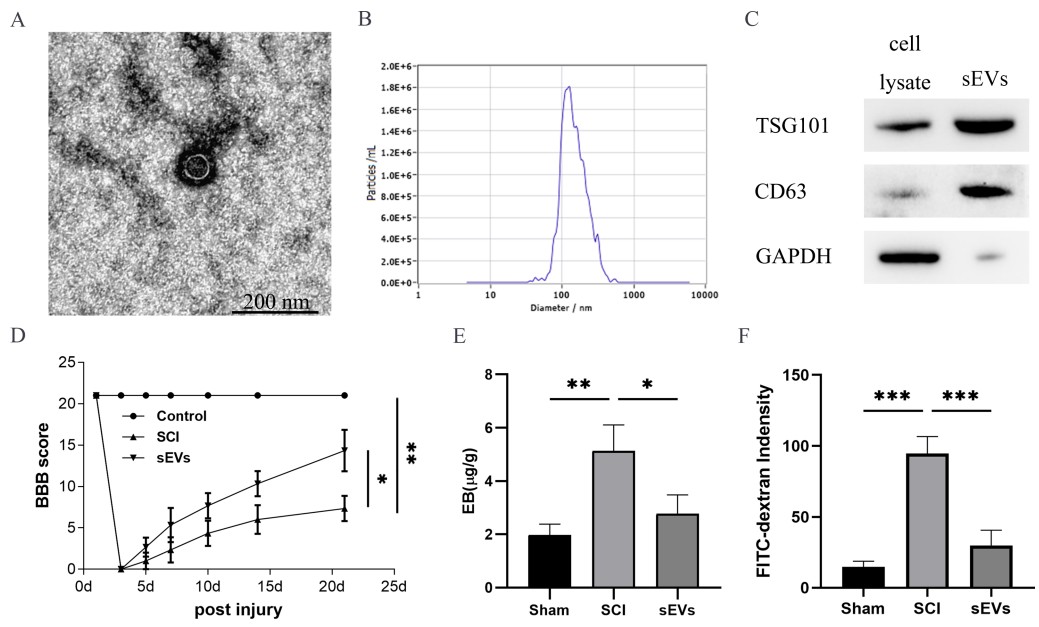

**Figure 1  hUC-MSCs-sEVs attenuate SCI-induced BSCB disruption.** (A) TEM photomicrographs of hUC-MSCs-sEVs; scale bar = 200 nm. (B) NTA results of hUC-MSCs-sEVs. (C) Western blotting showed the presence of exosomal markers, including CD63, and TSG101, in hUC-MSCs-sEVs. (D) The BBB scores. *$P < 0.05$ *versus* the Sham group; **$P < 0.01$ *versus* the Sham-operated group; $n = 3$. (E) Quantification of the amount of Evans Blue at 7 day ($\mu g/g$). (F) FITC-dextran was used in the spinal cord peripheral penetration analysis results at 7 day.

## Statistical analysis

All the experiments were performed three times at least. All data are shown as mean ± standard deviation, and statistical analysis was performed in GraphPad Prism (version 8.0, GraphPad Software Inc., USA). One-way ANOVA followed by Tukey's post hoc analysis was used for multiple comparisons. *P*-value < 0.05 was considered statistically significant.

## RESULTS

### hUC-MSCs-sEVs attenuate SCI-induced BSCB disruption

The successful isolation of the hUC-MSCs-sEVs was confirmed by TEM, which revealed their characteristic cup-shaped morphology (Fig. 1A). The hUC-MSCs-sEVs isolates were further identified using NTA, showing that particles with a diameter of 100 to 140 nm were the predominant populations (Fig. 1B). Western blot analysis of the sEVs lysates demonstrated significant positive bands for CD63 and TSG101, indicating the presence of exosomal markers, while GAPDH was employed as a control for purity (Fig. 1C).

The BBB scores were utilized to evaluate the functional recovery. Comparison of locomotor activity between the Exo group and the SCI Group 3–21 days post-injury revealed a remarkable improvement in locomotor function following sEVs therapy (Fig. 1D). Evaluation of BSCB integrity was performed using Evans blue and FITC-dextran fluorescence assays. hUC-MSCs-sEVs significantly reduced the fluorescence intensity of
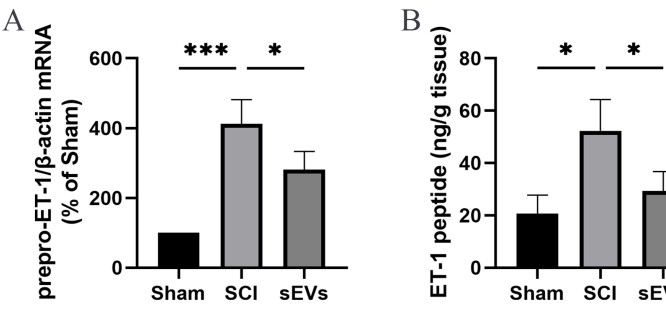

**Figure 2** **hUC-MSCs-sEVs increase SCI-induced ET-1 production.** (A) Increased prepro-ET-1 mRNA after SCI. Expression levels of prepro-ET-1 mRNA in rat spinal cord tissue were measured at 1 day after SCI. Expression levels of prepro-ET-1 mRNA were normalized to GAPDH. Results represent mean ± SEM. *$P < 0.05$ *versus* the Sham group; **$P < 0.01$ *versus* the Sham group; $n = 3$. (B) Increased ET-1 peptide after SCI. Production of ET-1 peptide in the spinal cord tissue was measured by ELISA at 1 day after SCI. The experiments were replicated three times. Results represent mean ± SEM, with experimental data shown as ET-1 peptide content (ng) per spinal cord tissue weight (g). *$P < 0.05$ *versus* the Sham group; $n = 3$. ***$P < 0.001$ *versus* the Sham group.

Evans blue in the injured spinal cord (Fig. 1E), and the penetration of FITC-dextran (Fig. 1F). Collectively, these findings demonstrate that hUC-MSCs-sEVs mitigate BSCB disruption in rats after SCI.

## hUC-MSCs-sEVs reduce SCI-induced ET-1 expression

Previous studies demonstrated an increase in ET-1 levels in the spinal cord tissue of SCI rats, suggesting that excessive ET-1 production following SCI contributes to vasoconstriction, which is closely associated with spinal cord ischemia and hypoxia symptoms (*A et al., 2019*). Therefore, we examined the effects of hUC-MSCs-sEVs on SCI-induced ET-1 production. Our results revealed a significant increase in the expression of prepro-ET-1 mRNA and ET-1 peptide following SCI. However, the administration of hUC-MSCs-sEVs at 200 ug/day after SCI reduced the SCI-induced upregulation of prepro-ET-1 mRNA (Fig. 2A) and ET-1 peptide (Fig. 2B).

## ET-1 is involved in the effects of hUC-MSCs-sEVs on SCI repair

To investigate the role of ET-1 in the cell neurological repair effects of hUC-MSCs-sEVs on the BSCB after SCI, ET-1 was administered at the injury site after SCI. Our results demonstrated that ET-1 injection significantly reduced the therapeutic effect of sEVs on motor activity 3-21 days post-SCI (Fig. 3A). Furthermore, at 24 h following SCI, Evans blue dye extravasation was assessed. The findings of the Evans blue dye (Fig. 3B) and Evans blue extravasation tests (Fig. 3C) revealed that ET-1 reversed the protective effect conferred by hUC-MSCs-sEVs. Additionally, FITC-dextran penetration, which was decreased by the administration of hUC-MSCs-sEVs, was significantly increased following ET-1 injection (Fig. 3D). According to the aforementioned data, hUC-MSCs-sEVs enhances functional recovery and lessens BSCB disruption following SCI *via* ET-1.

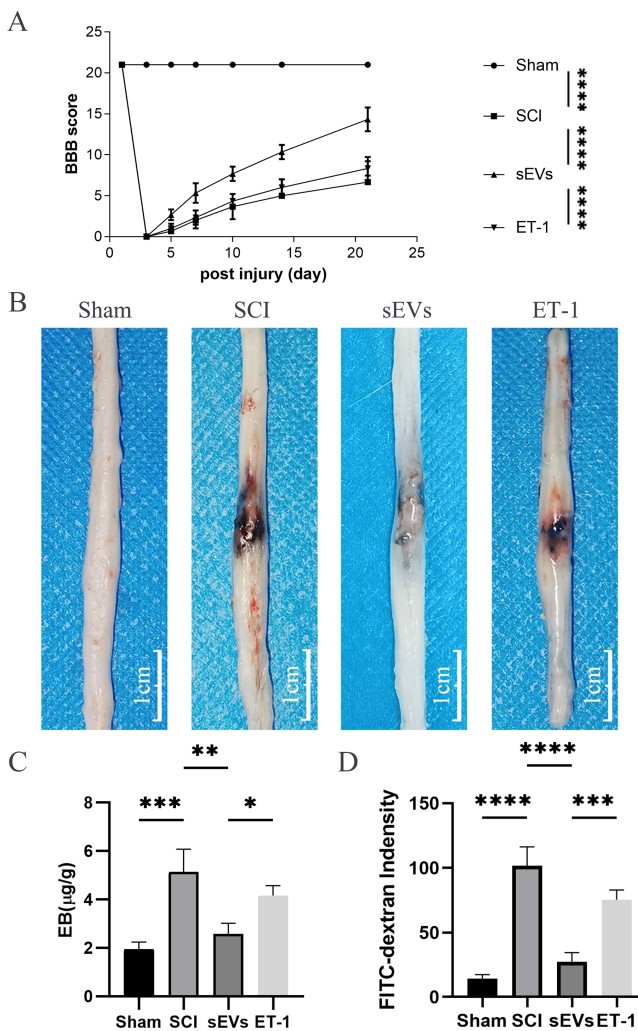

**Figure 3** **ET-1 is involved in the effects of hUC-MSCs-sEVs on SCI repair.** (A) The BBB scores. ****$P <$ 0.0001 *versus* the Sham group; $n = 3$. (B) Representative spinal cords show that Evans Blue dye permeabilized the injured spinal cord at 7 day; scale bar $= 3$ mm. (C) Quantification of the amount of Evans Blue at 7 day ($\mu g/g$). *$P < 0.05$ *versus* the Sham group; **$P < 0.01$ *versus* the Sham group; ***$P < 0.001$ *versus* the Sham group; $n = 3$. (D) FITC-dextran was used in the spinal cord peripheral penetration analysis results at 7 day. The experiments were replicated three times. ***$P < 0.001$ *versus* the Sham group; ****$P < 0.0001$ *versus* the Sham group; $n = 3$.

## hUC-MSCs-sEVs increase the expression of junction proteins after SCI by downregulation of ET-1

We performed a Western blot analysis to examine whether hUC-MSCs-sEVs can protect the integrity of the BSCB by regulating tight junction proteins and adhesion junction proteins. Our results demonstrated a significant reduction in the expression levels of ZO-1, β-catenin, occludin, and claudin-5 following SCI. However, treatment with hUC-MSCs-sEVs attenuated these changes, thereby promoting the restoration of BSCB integrity. Notably, the therapeutic effect of sEVs was also significantly compromised upon ET-1

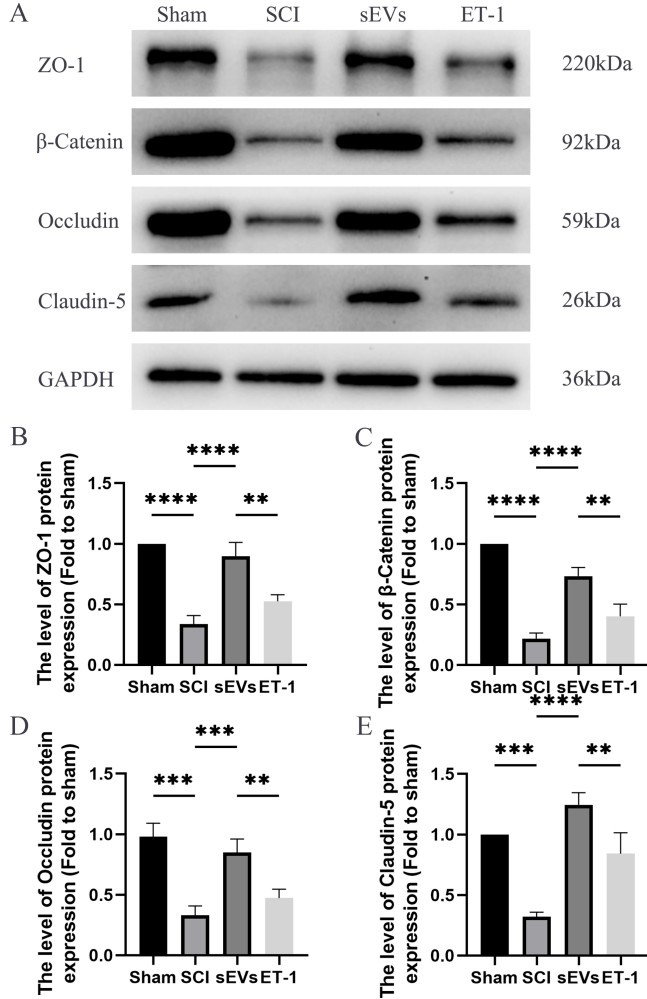

**Figure 4** hUC-MSCs-sEVs increase the expression of junction proteins after SCI by down-regulation of ET-1. (A–E) Western blot analysis of zo-1, β-catenin, occludin, and claudin-5 in the spinal cord of the Sham, SCI, sEVs, and ET-1 groups 1 day after SCI. The experiments were replicated three times. **$P <$ 0.01 *versus* the Sham group; ***$P < 0.001$ *versus* the Sham group; ****$P < 0.0001$ *versus* the Sham group; $n = 3$.

injection (Figs. 4A–4E). According to the aforementioned data, hUC-MSCs-sEVs enhances expression of cell junction proteins following SCI *via* downregulation of ET-1.

## hUC-MSCs-sEVs decreases expression of inflammatory mediators in SCI rats by downregulation of ET-1

Previous studies have highlighted the crucial role of matrix metalloproteinase (MMP) in the recovery process following SCI (*Yu et al., 2008*). Notably, MMP-2 and MMP-9 are known to be modulated by ET-1 (*He, Prasanna & Yorio, 2007*; *Wang et al., 2010*). Therefore, we subsequently evaluated the levels of MMP-2 and MMP-9 to confirm that the impact of hUC-MSCs-sEVs therapy in SCI rats was caused by the reduced expression of ET-1. Western blot analysis revealed significantly elevated expression levels of MMP-2

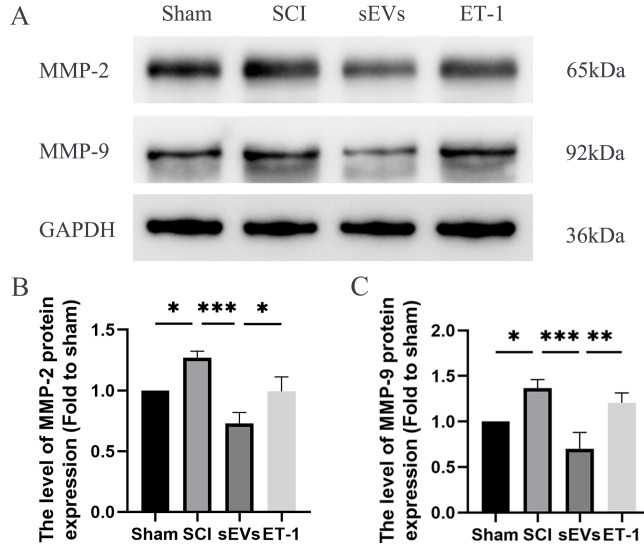

**Figure 5** **hUC-MSCs-sEVs decreases expression of inflammatory mediators in SCI rats by down-regulation of ET-1.** (A–C) Western blot analysis of MMP-2, and MMP-9 in the spinal cord of the Sham, SCI, sEVs, and ET-1 groups 1 day after SCI. The experiments were replicated three times. *$P < 0.05$ *versus* the Sham group; **$P < 0.01$ *versus* the Sham group; *** $P < 0.001$ *versus* the Sham group; $n = 3$.

and MMP-9 in SCI rats compared to the Sham group. However, the administration of hUC-MSCs-sEVs resulted in a significant decrease in MMP-2 and MMP-9 expression. The therapeutic effect of sEVs was reduced upon ET-1 injection (Figs. 5A–5C).

## hUC-MSCs-sEVs increase the expression of junction proteins in HBMECs after OGD/R by downregulation of ET-1

To detect the effects of hUC-MSCs-sEVs on OGD/R-injured HBMECs, we conducted a series of experiments, including Western blot analysis, Cell Viability Assay, and Paracellular Permeability Assay. The expressions of junctional proteins, including Claudin-5, Occludin, beta-Catenin, and ZO-1, were significantly decreased in HCMECs subjected to OGD/R. However, the presence of hUC-MSCs-sEVs notably reversed this decrease in expression, and this therapeutic effect was reduced upon ET-1 administration (Figs. 6A–6E). Moreover, we detected that hUC-MSCs-sEVs significantly enhanced cell viability, whereas ET-1 exerted an inhibitory effect (Fig. 6F). To investigate the impact of OGD/R and ET-1 on the integrity of HBMECs, FITC-dextran was added to the cells. Our results confirmed that OGD/R significantly increased cell permeability, while hUC-MSCs-sEVs addition significantly attenuated this effect. Interestingly, the introduction of ET-1 significantly increased endothelial barrier permeability (Fig. 6G).

## DISCUSSION

In this study, we demonstrated that hUC-MSCs-sEVs mitigate neurological impairments by preserving the integrity of the BSCB in SCI-affected rats. We demonstrated that hUC-MSCs-sEVs suppress ET-1 expression, thereby preventing the disruption of cell junctions

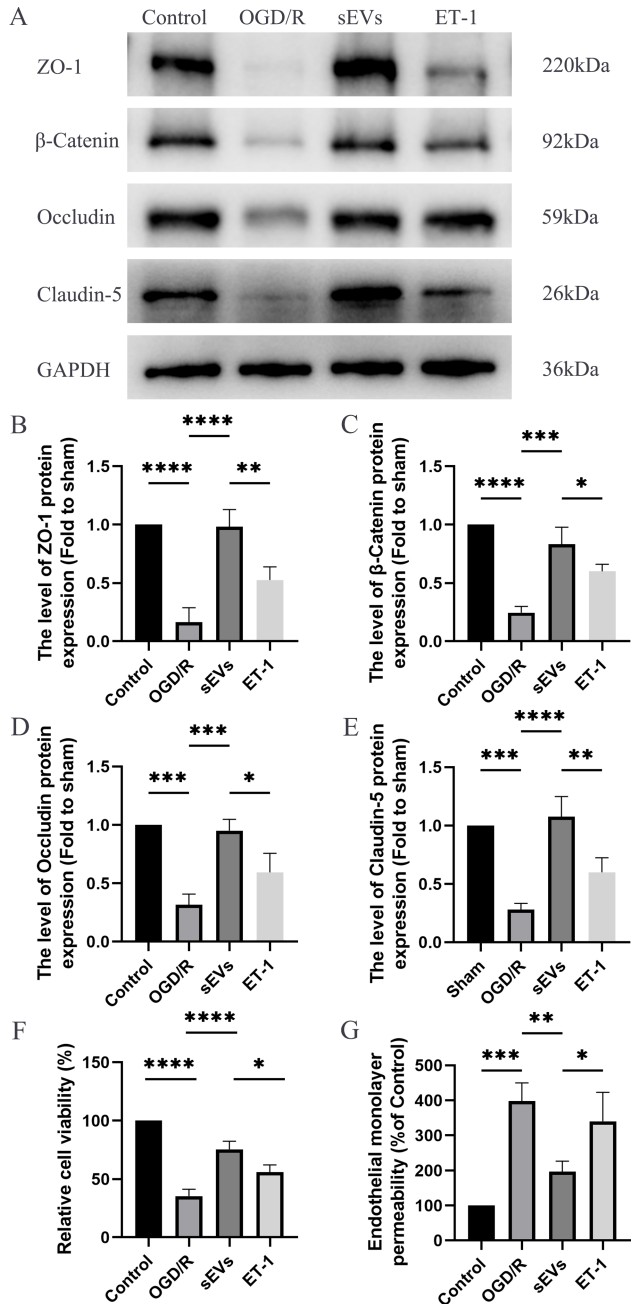

**Figure 6** **hUC-MSCs-sEVs increase the expression of junction proteins in endothelial cells after OGD/R by down-regulation of ET-1.** (A–E) Western blot analysis of zo-1, β-catenin, occludin, and claudin-5 in the HBMECs of the Control, OGD/R, sEVs, and ET-1 groups. (F) The viability of HBMECs of the Control, OGD/R, sEVs, and ET-1 groups was tested by CCK-8 analysis. The experiments were replicated three times. (G) Under different conditions, FITC-dextran permeates the fluorescence intensity of the lower chamber. $*P < 0.05$ *versus* the Control group; $**P < 0.01$ *versus* the Control group; $***P < 0.001$ *versus* the Control group; $****P < 0.0001$ *versus* the Control group; $n = 3$.

following SCI, facilitating BSCB repair. Our study sheds light on the underlying mechanism through which hUC-MSCs-sEVs exert their influence on BSCB integrity after SCI.

Growing evidence indicates that the BSCB may be indispensable to the pathophysiology of SCI (*Jin et al., 2021*). This barrier maintains the homeostasis of the spinal cord by regulating molecular exchanges between blood vessels and spinal parenchyma (*Abbott et al., 2010*). However, in clinical settings and animal models, SCI often leads to BSCB destruction, resulting in morphological and functional changes, such as vascular alterations, increased permeability, spinal cord edema, and spinal cord cavity formation (*Jin et al., 2021*). Mesenchymal stem cells derived sEVs have been shown to impact various processes, such as neuronal apoptosis, angiogenesis, and inflammation in SCI. (*Liu et al., 2019*) However, their role in BSCB repair remains an ongoing investigation. In our current study, we performed various assays for BSCB permeability, such as Evans Blue Dye Assays, FITC-Dextran Assays, and Paracellular Permeability Assay, in SCI-model rats, with or without sEVs treatment. Our findings demonstrate that hUC-MSCs-sEVs effectively reduce neurological impairments by preserving the integrity of the BSCB in rats with SCI.

Tight junctions between individual endothelial cells highly regulate the paracellular diffusion pathway in brain capillary endothelial cells (*Abbott et al., 2010*). Among the plasma membrane proteins responsible for the formation of tight junctions are claudin, occludin, and adherens junction molecules. The zonula occludens protein and cingulin form the cytoplasmic components of tight junctions (*Abbott, Rönnbäck & Hansson, 2006*; *Bernacki et al., 2008*). We examined tight junction membrane proteins and cytoplasmic components, such as claudin-5, Occluding, β-Catenin, and ZO-1, using Western blot analysis. Our results confirmed that hUC-MSCs-sEVs regulate the expression of tight junction proteins, thereby alleviating BSCB disruption after SCI.

Matrix metalloproteinases (MMPs) are a family of zinc-containing peptidases secreted by neutrophils. These peptidases destroy and restructure the extracellular matrix as well as other extracellular proteins. Matrix metalloproteinases are an essential component of barrier function (*Beck et al., 2010*). MMPs could instantly infiltrate the parenchyma of the spinal cord following the injury and continue to reside at the lesion site for more than 10 days (*Carlson et al., 1998*). MMPs, particularly MMP-2 and MMP-9, are prominently expressed 7 days after SCI and contribute to BSCB breakdown under pathological conditions (*Yao et al., 2018*; *Wang et al., 2021*). Our study also revealed elevated expression of MMP-2 and MMP-9 in the spinal cord tissue of rats after SCI. Administration of hUC-MSCs-sEVs decreased the expression of MMP-2 and MMP-9, indicating that hUC-MSCs-sEVs could mitigate BSCB disruption mediated by these MMPs. Our findings also provide further insight into the molecular mechanisms underlying the protective effects of sEVs on BSCB disruption after SCI.

Notably, our results show that the beneficial effects of hUC-MSCs-sEVs on SCI were significantly inhibited in the presence of ET-1, a vasoconstrictive peptide composed of 21 amino acids. Increased expression of ET-1 has been correlated with the pattern of BSCB degradation after SCI. Pharmacological blockade of ET-1–mediated vasoconstriction has been shown to attenuate BSCB degradation after SCI (*McKenzie et al., 1995*). Intrathecal administration of ET-1 in the intact spinal cord resulted in disruption of the BSCB

(*Westmark et al., 1995*). In this study, ET-1 injection in the wound site significantly reduced the therapeutic effects of hUC-MSCs-sEVs. *In vitro* experiments further demonstrated that hUC-MSCs-sEVs increased the expression of tight junction and adhesion junction proteins, enhanced cell viability, and decreased cell permeability following OGD/R. Furthermore, the administration of ET-1 to HBMECs led to corresponding changes in the expression of tight junction proteins, adhesion junction proteins, cell viability, and permeability. These findings confirm that the regulatory effect of hUC-MSCs-sEVs on BSCB function is mediated through the modulation of ET-1 expression.

It should be noted that there are some limitations to this study. First, sEVs consist of a class of nanovesicles that contain various substances derived from parental cells, such as miRNA, mRNA, and protein, and transport them to recipient cells (*Kalluri & LeBleu, 2020*). Exactly which of these components exerted a role in regulating ET-1 and ameliorating BSCB destruction was not investigated in this study. These functional components need to be further identified in subsequent studies. Second, the spinal cord has a variety of cell types, each of which performs a distinctive function and interacts with others in the destruction of the BSCB after SCI. Due of this, microglia and astrocytes both contribute significantly to the development of BSCB destruction following SCI. In the current investigation, we only looked at how sEVs affected endothelial cells. It is important to carry out additional *in vitro* and *in vivo* research using co-culture systems with mixed endothelial cells and other spinal cord cell types. Third, the current study did not examine ET-1 receptors ETaR and ETbR or additional signalling pathways that are involved in ET-1 intracellular signalling. Future research will need to focus on these receptors and signalling pathways. Forth, only female rats were chosen for this study, because they are more commonly used in current spinal cord injury research and they exhibit less fighting behavior compared to males, which facilitates comparisons and references to previous studies and improves the stability of experimental results (*Cheng et al., 2012*; *Torres-Espín et al., 2018*; *Vawda et al., 2019*). In our future research, we consider including male models to obtain a more comprehensive understanding of the regulatory mechanisms of the therapeutic effects of hUC-MSCs-sEVs on SCI.

## CONCLUSIONS

In conclusion, the current study provides evidence for the protective effect of hUC-MSCs-sEVs on BSCB integrity following SCI. We demonstrate that hUC-MSCs-sEVs attenuate BSCB degradation and promote functional recovery after SCI by regulating the expression of ET-1. We further elucidate the mechanism by which hUC-MSCs-sEVs exert their influence on BSCB integrity after SCI.

## ACKNOWLEDGEMENTS

We would like to thank the Shanxi Provincial Key Laboratory of Kidney Disease, the Core Laboratory at Shanxi Provincial People's Hospital and the Laboratory Animal Center at Shanxi Province Cancer Hospital for their technical support. We thank Bullet Edits Limited for the linguistic editing and proofreading of the manuscript.

### Funding

This work was supported by the Natural Science Foundation of Shanxi Province (201901D111410) and the Outstanding Youth Foundation of Shanxi Bethune Hospital (2019YJ07). The funders had no role in study design, data collection and analysis, decision to publish, or preparation of the manuscript.

### Grant Disclosures

The following grant information was disclosed by the authors:
Natural Science Foundation of Shanxi Province: 201901D111410.
Outstanding Youth Foundation of Shanxi Bethune Hospital: 2019YJ07.

### Competing Interests

The authors declare there are no competing interests.

### Author Contributions

- Chenhui Xue conceived and designed the experiments, performed the experiments, analyzed the data, prepared figures and/or tables, authored or reviewed drafts of the article, and approved the final draft.
- Xun Ma conceived and designed the experiments, authored or reviewed drafts of the article, and approved the final draft.
- Xiaoming Guan conceived and designed the experiments, authored or reviewed drafts of the article, and approved the final draft.
- Haoyu Feng conceived and designed the experiments, authored or reviewed drafts of the article, and approved the final draft.
- Mingkui Zheng performed the experiments, authored or reviewed drafts of the article, and approved the final draft.
- Xihua Yang conceived and designed the experiments, authored or reviewed drafts of the article, and approved the final draft.

### Animal Ethics

The following information was supplied relating to ethical approvals (i.e., approving body and any reference numbers):

Shanxi Provincial People's Hospital Institutional Animal Care and Use Committee provided full approval for this research (89/2022).

### Data Availability

The raw measurements are available in the Supplementary Files.

### Supplemental Information

Supplemental information for this article can be found online at http://dx.doi.org/10.7717/peerj.16311#supplemental-information.

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
