# Peer review of "Small extracellular vesicles derived from umbilical cord mesenchymal stem cells repair blood-spinal cord barrier disruption after spinal cord injury through down-regulation of Endothelin-1 in rats"

_PeerJ, doi:10.7717/peerj.16311_

## Round 0.1 · original submission · Major Revisions

Both reviewers provide clear indication of what they consider to be the most pressing issues. I add my own here.

(i) https://www.isev.org/rigor-standardization seeks to standardise the methodological approaches used in exosome isolation. You should cite this web site and explain how closely your method fits (or does not) to their recommended approaches. This is regarded as essential information.

(ii) the issue of cell passage is clearly important, and an attempt must be made to show whether there is variation in exosome content/efficacy with passage number. This is regarded as essential information.

(iii) for every set of experimental data, the number of biological and technical replicates must be clearly stated. This is regarded as essential information.

(iv) only one set of immunoblots is presented for each figure yet the authors indicate many replicates were performed. All replicate data must be uploaded, suitably labelled, as supplemental data.

(v) the raw statistical data file is inadequate. All technical and biological replicates must be presented and the data needs clearer labelling to allow fuller evaluation.

(vi) the section explaining why only female rats were used must provide either a compelling rationale, or be modified to include all biological sexes. One interpretation is that the effect is absent in males? This is regarded as essential information.

(vii) clear data establishing the validity of the endothelial and glial cells is required (e.g. marker immunoblots) to provide certainty. This is regarded as essential information.

(viii) the TEM data in figure 1 is inadequate. Clearer and more sharper images are needed. This is regarded as essential information.

I will expect all of these points to be fully and completely addressed in any revision.

I hope these comments are of use.

·

Basic reporting

Literature references, sufficient field background/context provided.

Professional article structure, figures, tables. Raw data shared

Self-contained with relevant results to hypotheses.

Experimental design

1) It is recommended that authors include the specific passage number for the cells utilised. This issue becomes more significant when dealing with HUC-MSC cells, as their plasticity diminishes beyond the fifth passage.
2) What is the rationale for selecting female rats over male rats in this study? The topic at hand warrants further discussion.
3) It is advised by the worldwide recommendation that authors should consider revising the term "exosomes" to "small extracellular vesicles (sEVs)."
4) The indication of exosome storage is necessary.
5) How was the concentration of 200 ug/mL for exosomes determined? This should be discussed

Validity of the findings

1) Inquiring about the presence of two lanes in Figure 1 C, what is the rationale behind this duplication in the Western blot (WB) results? Are there variations among different sets of exosomes? Is the first lane representative of cells, while the second lane represents exosomes?
2) The suggested revision for line 255 is to modify the headline to "HUC-MSCs-Exos reduce SCI-induced ET-1 expression" since the exosomes were shown to decrease its expression, rather than enhance it.
3) According to Figure 3B, It is advisable to position a ruler close to the photographs. Nevertheless, this action was not executed previously. It is recommended to include a virtual ruler adjacent to the photographs to facilitate the estimation of their length.
4) Figure 4A is depicted in the following manner. It is imperative to provide the molecular weight of the protein.

Additional comments

The introduction has the potential to be effectively crafted to establish coherent connections between topics. The present composition consists of paragraphs that are independent entities and lack cohesive connections between them. For instance, while authors have acknowledged the potential hazards associated with stem cells, they have not provided an explanation as to why exosomes outperform cell therapy. Moreover, the use of stem cells remains unaddressed.

·

Basic reporting

Overall, the paper provides a comprehensive analysis of the effects of uHC-MSCs-derived exosomes in spinal cord injury. The authors present their findings in a clear and organized manner. There are a few suggestions below:

For Introduction

1. Author acknowledges that cell-based therapies and stem cell transplantation carry potential risks, but the rationale for the use of exosome over cell-based therapies and stem cell transplantation is not provided. Suggest highlighting the issue/problems with current standard treatment for spinal cord injury and link it with the potential use of stem cell exosomes.

2. There appears to be a lack of coherence which hinders the seamless integration of ideas. The introduction would be beneficial by establishing more robust links between paragraphs. Consider adding transitional phrases or providing a clearer roadmap at the beginning to help readers understand how each idea connects to the next.

Experimental design

Materials and Methods

1. Under subsection “cell culture”, suggest indicating the passage number of hUC-MSCs used to isolate exosomes because exosomes isolated from a higher passage number can lead to senescence or other changes in stem cells, affecting exosomes secretion and overall efficacy. This could potentially affect the optimal quality and therapeutic potential of exosomes in medical treatment.

2. Under subsection “Treatment”, the author highlighted the concentration of exosomes injected to the treatment group as 200 μL exosomes solution (200 μg/mL), suggest to indicate the amount of exosomes injected as 40 μg exosomes (200 μg/mL) to help the reader understand the exact amount of exosomes injected.

3. Recommend standardizing the use of microlitres (μL) with a capital ‘L’ to ensure consistency and clarity throughout the paper.

Validity of the findings

Results

1. Figure 1C, suggest labeling the 2 lanes.

2. Study showed that administration of hUC-MSCs-Exos at 200 μg/day after SCI reduced prepro-ET-1 mRNA and ET-1 peptide. According to Figure 2A and 2B, prepro-ET-1 mRNA and ET-1 peptide for Exos-treated group are lower compared to SCI group but higher compared to Sham group. Suggest to revise title for subsection “hUC-MSCs-Exos increase SCI-induced ET-1 production”.

Additional comments

No comment

---

## Round 0.2 · Minor Revisions

Thank you for addressing the issues raised clearly and effectively. Two points remain, as detailed by the review included. Please carefully and clearly address these issues in a revised paper.

·

Basic reporting

no comment

Experimental design

Most of the comments I provided have been addressed by the authors. I still have some remaining problems that require attention.

I previously inquired as to why a female model was included in the study, but not a male model. The authors have omitted the term "female" from the paper without offering any accompanying rationale. Indeed, the data supplied appears to have remained unchanged. In this particular instance, it is presumed that the male model was still omitted. It is imperative for authors to present a rationale for the inclusion of just female models, while excluding male models, in their research. This justification should be thoroughly explained within the discussion section. Excluding the term "female" from the text does not imply the inclusion of all gentles.

In a previous inquiry, I sought information regarding the administration of 200 ug/mL or 40 ug of small extracellular vesicles (sEVs). I was examining the process by which this dosage was determined for the treatment. Is there any evidence from cell culture or past studies suggesting that this dosage is the effective dose? This should be discussed.

Validity of the findings

no comment

---

## Round 0.3 · accepted · Accept

Thank you for addressing these remaining points. I am happy to accept the paper now.